# PrimPol: A Breakthrough among DNA Replication Enzymes and a Potential New Target for Cancer Therapy

**DOI:** 10.3390/biom12020248

**Published:** 2022-02-03

**Authors:** Alberto Díaz-Talavera, Cristina Montero-Conde, Luis Javier Leandro-García, Mercedes Robledo

**Affiliations:** 1Hereditary Endocrine Cancer Group, Human Cancer Genetics Program, Spanish National Cancer Research Centre (CNIO), 28029 Madrid, Spain; cmontero@cnio.es (C.M.-C.); ljleandro@cnio.es (L.J.L.-G.); mrobledo@cnio.es (M.R.); 2Centro de Investigación Biomédica en Red de Enfermedades Raras (CIBERER), Instituto de Salud Carlos III (ISCIII), 28029 Madrid, Spain

**Keywords:** PrimPol, primase, polymerase, repriming, replication stress, genomic instability, cancer

## Abstract

DNA replication can encounter blocking obstacles, leading to replication stress and genome instability. There are several mechanisms for evading this blockade. One mechanism consists of repriming ahead of the obstacles, creating a new starting point; in humans, PrimPol is responsible for carrying out this task. PrimPol is a primase that operates in both the nucleus and mitochondria. In contrast with conventional primases, PrimPol is a DNA primase able to initiate DNA synthesis de novo using deoxynucleotides, discriminating against ribonucleotides. In vitro, PrimPol can act as a DNA primase, elongating primers that PrimPol itself sythesizes, or as translesion synthesis (TLS) DNA polymerase, elongating pre-existing primers across lesions. However, the lack of evidence for PrimPol polymerase activity in vivo suggests that PrimPol only acts as a DNA primase. Here, we provide a comprehensive review of human PrimPol covering its biochemical properties and structure, in vivo function and regulation, and the processes that take place to fill the gap-containing lesion that PrimPol leaves behind. Finally, we explore the available data on human PrimPol expression in different tissues in physiological conditions and its role in cancer.

## 1. Introduction: PrimPol, a New Enzyme Working on DNA Replication

In human cells, DNA replication is initiated by the primosome [1,2]. This latter consists of a primase (composed of catalytic and regulatory subunits) and a DNA polymerase α (also composed of catalytic and regulatory subunits). Primases start the synthesis by catalyzing the formation of RNA primers, which are 8–9 nucleotides in length in both strands (Figure 1A). This limited length is due to the continued interaction of the first 5‘ ribonucleotide with the regulatory subunit, during the formation of the full primer and its inherent conformational change; once the RNA primer is an 8/9-mer, it is transferred to the other member of the primosome, Pol α (Figure 1B). This DNA polymerase is able to elongate the small RNA primers with dNTPs, making longer and hybrid primers of 30 nucleotides with a 5′ portion of RNA and a 3′ portion of DNA (Figure 1C) [3,4,5]. This step is required for the replicative DNA polymerases, which prefer to elongate DNA primers over RNA primers (Figure 1D). DNA replication enzymes only synthetize in the 5′–3′ direction, which implies an inherent asymmetry of this process. The semi-discontinuous model of DNA replication [6,7] establishes that the leading-strand requires the priming event for the primosome and elongation for the replicative DNA polymerase Pol ε (Figure 1A–D) [8,9]. However, the lagging-strand is synthesized discontinuously in small pieces, which are initiated by the primosome (Figure 1E) and elongated by the replicative DNA polymerase Pol δ (Figure 1F). Therefore, the primases, Pol α and Pol δ, are recurrently involved in lagging-strand synthesis [9,10]. The fragments synthetized from the lagging-strand are called Okazaki fragments (Figure 1F,G) [11], and are subsequently joined to complete a duplex DNA.

For many years, it has been accepted that the DNA replication of the leading strand only required one unique and single priming event performed by the primosome [6]. This is formally true in the absence of obstacles in the elongation step by the replicative DNA polymerase Pol ε. However, when obstacles arise and Pol ε stalls, a solution is required to continue synthesis, repriming being one of these solutions. In human cells, this latter is carried out by a single and specific DNA primase called PrimPol [12,13,14,15], an enzyme discovered in 2013 that provides a more “ergonomic” solution compared with the primosome.

## 2. Similarities and Differences among Polymerizing Enzymes of DNA Replication: RNA Primases, DNA Polymerases, and PrimPol

RNA Primases, DNA polymerases, and PrimPol are enzymes that catalyze the same chemical reaction: the nucleotidyl transferase reaction using a two-metal ion mechanism [16]. This reaction is a nucleophilic substitution type 2 (S_N_2), in which the 3′OH group attacks the α phosphorus of the incoming nucleotide, releasing a pyrophosphate (phosphates β and γ of the incoming 3′ nucleotide) (Figure 2A,B) [17,18,19,20,21]. Nevertheless, these three enzymes have the following particularities.

### 2.1. RNA Primases

RNA primases (classically called DNA primases) are classified in two major groups [22,23]. One group comprises bacterial and phage primases with a structural similarity to *E. coli* DnaG primase; a second group comprises archaeal and eukaryotic primases (AEP), which also includes the primases of viruses. Both groups differ in their structure and in the relationship with other proteins at the replication fork. DnaG-like primases are associated with the helicase, whereas canonical AEPs are associated with Pol α, as described above for human cells. Nevertheless, both groups of primases perform a de novo synthesis of small pieces of RNA. These RNA primers are initiated by the binding of two nucleoside triphosphates that are selected by two bases in the ssDNA template, and then a dimer is formed by the general nucleotidyl transferase reaction. Primases can use a single NTP as the molecule that provides the attacking 3′OH group as these enzymes possess a pocket, which contains a highly positive charge, to harbor the triphosphate of the nucleotide at the 5′ site (the priming site) [3,5] (Figure 2A). The efficiency of this initiating reaction is quite low in comparison to normal polymerization [24]. To facilitate this limiting step, primases recognize a neighboring base adjacent to the base that selects the 5′ nucleotide, which is called the cryptic base (Figure 2A). In the case of the human RNA primase p49, the specific Histidine^303^ interacts with the cryptic base making a “fake base pair” that stabilizes the binding and selection of the 5′-nucleotide [3,5]. Moreover, these canonical RNA primases improve their efficiency by using NTPs since they are a highly abundant substrate in cells compared to dNTPs [22,23,25]. As explained above, the action of RNA primases takes place at the beginning of replication on both leading and lagging strands and in a recurrent manner on lagging strands for each Okazaki fragment, being fundamental for initiating the synthesis. RNA primers catalyzed by canonical RNA primases can be easily and accurately removed based on their RNA nature in further replication steps [26,27,28].

### 2.2. Polymerases

Conversely to RNA primases, DNA polymerases are not able to initiate de novo DNA synthesis [29], as they need an elongating starting point different to the single nucleotide that provides the first attacking 3′OH group. Therefore, DNA polymerases are only able to catalyze elongation with dNTPS from DNA or RNA primers. Mostly, this 3′OH group is provided to DNA polymerases by RNA primers made by RNA primases [3,5,30,31], but also by specific amino acids in the so-called “priming proteins” [32]. This group can also be provided by a DNA end, as is the case in the nicks and gaps [33] or double-strand breaks [34] (Figure 2B). This requirement is explained by the lack of polymerases in the positive-charged pocket, which harbor the triphosphate of the nucleotide at the 5′ site, impeding the binding of these substrates at this position. 

DNA polymerases, especially in the case of those that carry out the synthesis of the bulk of DNA, discriminate in an efficient manner against the incorporation of NTPs due to their potential hazards, as NMPs embedded in DNA can alter the helix and impair its proper functions [35,36]. The structural basis for this discrimination is called “steric gates”, which are usually formed by residues with bulky side chains, such as tryptophan or tyrosine, which impede the entrance of NTPs into the active site by steric clashing with the 2′ hydroxyl group (2′OH) [37,38] (depicted in Figure 2B). Nevertheless, even considering the efficiency of this step, the exclusive use of dNTPs by DNA polymerases is not guaranteed, mainly due to the low abundance in cells of dNTPs compared to NTPs [39]. Recent studies demonstrate that replicases, which are highly efficient in sugar discrimination in vitro, incorporate NTPs at high rates in vivo (e.g., in the case of yeast Pol ε, around 1 per 1 kb) [40]. In fact, it was recently determined that, in general, ribonucleotides are the most common incorrect nucleotides incorporated in DNA [41].

DNA polymerases are classified by their structure into different families. In human cells, there are 16 different DNA polymerases that perform different functions, mainly defined by their active site. Those who carry out the replicative task have a tight active site, which allows only the base pair C:G or A:T to be harbored (Watson and Crick base pairs). This fact implicates that replicative DNA polymerases perform a high-fidelity synthesis since they only allow correct base pairing, rendering them unable to carry out non-canonical reactions [29]. Replicative DNA polymerases such as Pol ε (leading-strand) and Pol δ (lagging-strand) are highly accurate at copying DNA. In addition to a high insertion fidelity, replicases have proofreading abilities, based on an evolutionarily conserved 3′-5′ exonuclease activity [42]. In human cells, the primosome component Pol α is the exception of a replicative DNA polymerase with a lower fidelity, which lacks the proofreading [43] that is apparently needed for its function in extending the RNA primers made by the primase, since the proofreading ability could erase the primers before its elongation [44]. In any case, its insertion fidelity is enough for its limited participation in elongation; moreover, it was shown that errors eventually made by Pol α may be removed by 3′-5′exonuclease activity of Pol δ when the synthesis is continued [45] or by the DNA mismatch repair pathway [46,47]. Moreover, Pol α DNA initiators are partially removed during the Okazaki fragment maturation, a process in which the strand displacement activity of Pol δ [48,49] is involved. Other proteins such as FEN1, DNA2 and DNA ligase [49,50,51] also participate. Considering all three of these processes, it is estimated that the final contribution of Pol α in DNA synthesis is 1.5% of the genome [52]. Mutations that alter the nucleotide misincorporation or proofreading abilities of Pol ε or Pol δ have been described in multiple cancer types [53,54,55]. Accordingly, a genetically engineered mouse model with a deficient Pol ε proofreading function, suffers an increase in mutation burden and cancer incidence [56]. 

DNA polymerases that are able to cause trans-lesion synthesis (TLS) have a laxer active site. TLS DNA polymerases, who normally carry out non-replicative functions [57], are capable of carrying out this task, but with the detriment of having a more error-prone action than the replicative polymerases; due to their open active sites, they can also harbor non-Watson and Crick base pairs. Thus, once TLS polymerases make their contribution tolerating the DNA damage, they must finish their function and be replaced by replicative DNA polymerases in order to avoid higher rates of mutagenesis [29,58]. Human cells have several TLS polymerases such as the Y-family members Pol η [59,60], Pol ι [61], Pol κ [62,63] and Rev1 [64,65,66], the B-family member Pol ζ [67], X-family members Pol λ [68], Pol μ [69,70] and A-family member Pol θ [71]. 

### 2.3. PrimPol

The most recently discovered polymerizing enzyme of human DNA replication is PrimPol, a novel primase-polymerase that belongs to the archaea–eukaryotic primases (AEP) superfamily and localizes in both mitochondria and the nucleus of human cells [12,72]. PrimPol is the first DNA primase characterized in human cells [12,72]; whereas the common primases start the synthesis using ribonucleotides in both priming and elongation site (5′ and 3′ sites, respectively), PrimPol is able to start the synthesis using dNTPs at the elongation site discriminating against NTPs. In vitro, PrimPol can use both NTPs and dNTPs at the 5′ nucleotide binding site (priming site); therefore, it is very likely that an NTP initiates PrimPol-made primers in vivo due to the abundance of NTPs. Conversely, the second and next nucleotides of the primer made by PrimPol are made from dNTPs (Figure 3A). Moreover, in vitro, PrimPol is able to elongate pre-existing primers with dNTPs acting as DNA polymerase (Figure 3B). Recently published research determined that, when acting as DNA polymerase, PrimPol is also able to carry out a gap filling reaction in vitro and possesses strand displacement activity [73]. 

PrimPol possesses a steric gate that discriminates against NTPs at the elongation (3′site) site in both primase and polymerase activities, the same sugar discriminator element used by DNA polymerases and explained earlier [74]. The residue that determines this PrimPol trait is Tyr^100^ (Figure 4, left panel), which has been found to mutate in cancer. Notably, cancer-associated missense mutation Y100H unleashes PrimPol ability to use NTPs at the elongation site by dismantling their steric gate (Figure 4, right panel) [74]. Moreover, Y100H over-expression in PrimPol-deficient cells leads to an increased tolerance to depletion of the dNTP pool levels in the S-phase using hydroxyurea [74]; this scenario is proposed as a model for the early stages of tumorigenesis [75]. 

PrimPol is also able to act as a TLS polymerase by directly reading lesions (Figure 3B) such as 8-Oxo-2′-deoxyguanosine (8oxodG), which can be copied both with dCTP (error-free) or with dATP (error-prone) [12,74,76,77,78,79,80], O6-methylguanine (O6-me-G) [78], 5-formyluracil (fU) [78], 5-Methyl-2′-deoxycytidine (mC) [81], 5-hydroxymethyl-2′-cytidine (hmC) [81] and 1,2-intrastrand cisplatin cross-link (1,2-GG CisPt CL) lesions [82]. In addition, PrimPol is capable of acting as TLS polymerase by realigning primers ahead of lesions that cannot be read directly by this enzyme (Figure 3B), such as abasic sites (AP) [12,76,78] and the UV-induced lesions: cyclobutane pyrimidine dimers (CPD; T = T) and (6–4) pp photoproduct [13,76]. Nevertheless, these TLS DNA polymerase capabilities have only been demonstrated in vitro. However, the PrimPol ability to prime downstream of a lesion has only been demonstrated in vivo, which makes this enzyme the first DNA primase with this function to be discovered [13]. PrimPol is able to catalyze the synthesis of de novo primer downstream of a readable lesion or an unreadable lesion (Figure 3A). Considering the in vitro abilities of PrimPol, it is probable that in vivo PrimPol uses its abilities to be proficient as a DNA primase involved in restarting stalled replication forks. In summary, PrimPol is a primase able to catalyze long DNA primers ahead of lesions with its regular DNA primase activity. Yet, only in vitro, PrimPol is able to use its TLS capability when polymerizing in two different ways: on the one hand, by directly reading lesions, such as 8oxodG; and, on the other hand, by skipping non-readable lesions, such as AP sites, realigning the nascent primer ahead to continue the synthesis (Figure 3B). 

Traditionally, it has been speculated that the reason why PrimPol catalyzes the synthesis of DNA primers could be easily explained by the fact that DNA primers are suitable substrates for replicative DNA polymerases [74]. The fact that RNA primers catalyzed by RNA primases are easily eliminated, minimizing mutagenesis during replication, while DNA primers catalyzed by PrimPol, could be more difficult to be erased is also a matter of discussion [74]. Thus, it has been suggested that this special feature of PrimPol is required for its TLS polymerase activity [74].

## 3. PrimPol: A New Player Alleviating Replication Stress

DNA replication eventually faces obstacles that can block the replication fork, such as base damages in the template, non-canonical structures or low availability of dNTPs. This leads to replication stress, a major source of genome instability [83,84]. To minimize the impact of replication stress, the DNA replication process has several ways to tolerate these problems. Among all of the pathways contributing to alleviate DNA replication stalling, the simplest mechanism is repriming ahead of the obstacle, providing a new starting point to the fork and allowing the replication to continue on its path [85,86,87]. Frequent repriming naturally occurs on the lagging strand. The obstacles on this strand could compromise the completion of a given Okazaki fragment, but not the synthesis of the recurrent new primers that are coupled to the helicase action and fork advance [88] (Figure 5A). In other words, it does not represent a setback that a given Okazaki fragment is blocked by an obstacle, because another new Okazaki fragment is initiated ahead of the lesion. Therefore, obstacles on the lagging strand have minimal impact in replication fork progression.

On the leading strand, the intrinsic continuity of its replication requires the unscheduled activation of different mechanisms when an obstacle is found. One of the mechanisms used to alleviate replisome stalling is fork reversal (FR) [89], catalyzed by several enzymes such as ZRANB3, HLTF, SMARCAL1, FBH1, RECQ1, RECQ5, BLM, WRN and FANCM [89,90]. FR is characterized by the so-called “chicken foot” structure: a backwards movement of the newly synthesized lagging strand, which melts from its template to serve as an alternative template for the blocked leading strand, resuming fork progression (Figure 5B). This pathway could also stabilize and protect the replication fork until other mechanisms are activated to complete the synthesis of this DNA region, i.e., activating a proximal replication origin [89,91], interstrand-crosslink (ICL) repair [92] or template switch [91]. Nevertheless, several aspects about the enzymology and the mechanism of fork reversal in vivo remain unknown. Anther mechanism used to alleviate replisome stalling is repriming [85,88]. In human cells, PrimPol is the enzyme specialized in repriming at stalled forks (Figure 5B), both in the nucleus [12,13,72] and mitochondria [12,93,94,95]. In vivo, PrimPol uses its DNA primase activity to mediate a fork restart in response to different replication obstacles such as UV lesions [13,96], G-quadruplexes (Figure 3A) [97], chain terminating nucleotides [96] or R-loops [98]. Moreover, PrimPol promotes resistance to several genotoxic external agents such as hydroxyurea (HU) [13,15,74,96], methylmethane sulfonate (MMS) [96] tenofovir [99], enzo[a]pyrene diol epoxide (BPDE) [100,101], cisplatin [96,102], or mitomycin C (MMC) [102]. Nevertheless, although PrimPol-mediated repriming mechanisms on the leading strand are very suitable mechanisms for a fork restart, the enzymology and mechanics of this proposed process are not yet well understood. Cisplatin and MMC are agents responsible for generating DNA ICLs. PrimPol also facilitates replication traverse of ICLs by priming downstream of the lesion [102] (Figure 6). Several aspects about PrimPol-mediated replication traverse remain unknown. Furthermore, PrimPol repriming is shown to be highly relevant in DNA replication even in the absence of external disturbances [13], likely due to the endogenous obstacles encountered in the path of DNA replication. The pathway choice in cells between fork reversal and repriming is under study. In human BRCA1-deficient cells, the most frequent event to deal with single cisplatin dose treatment is fork reversal, whereas this mechanism is replaced by PrimPol repriming as an adaptive response to treatment with multiple doses of cisplatin [103] (Figure 5B). In addition to this, the suppression of fork reversal triggers unrestrained DNA replication, partially mediated by PrimPol [104], and promotes more repriming [105]. Altogether, these data reveal that human cells prioritize fork reversal over repriming as the initial response to DNA replication disturbance. However, PrimPol repriming activity is key as an alternative mechanism to maintain DNA replication in certain circumstances, such as the cumulative ICL induced by multiple cisplatin dose treatments. The fine-tuned understanding of adaptive responses to DNA damage clearly hold a therapeutic value, as they may lead to synthetic lethality strategies for cisplatin-resistant cancer cells.

## 4. Structure and Regulation of Human PrimPol

Human PrimPol is composed of an AEP core that contains a single active site at the N-terminal of the protein and a Zn finger [12,72], as well as an RPA binding domain (RBD) at the C-terminal region (Figure 7A) [106]. A solution to the AEP core structure, in complex with a DNA template/primer and incoming dNTP, was found in 2016 [107]. The active site of PrimPol has the following conserved key residues (compiled in Figure 7A and Table 1) that have been studied in vitro: the three conserved carboxylates D^114^, E^116^, and D^280^, which are the residues that cause the covalent binding of the two-metal cations used to carry out the above mentioned nucleotidyl transferase reaction [12,79,107]. Mn^2+^ is the preferred metal for PrimPol activities in vitro [12,77,79,108,109,110], despite having lesser fidelity than with Mg^2+^ [77,109,110]. Mutations of the three carboxylates for alanine erase primase and polymerase activities [12,79]. Interestingly, E^116^ plays a crucial role in favoring the use of Mn^2+^ since the mutation E116D of PrimPol impairs this ability [79]. H^169^ [12,107] and R^291^ [107,111] are the key residues needed to stabilize the incoming nucleotide. Mutation of R^291^ [111] to alanine highly impair primase and polymerase activities. Y^100^ plays the abovementioned role as a “steric gate”, impeding the entrance of ribonucleotides in the 3′position by clashing with their 2′-OH groups [74]. R^47^ and R^76^ contact the DNA template, and their mutations to alanine highly impair PrimPol activities [107,112]. The C-terminus of the protein, the structure of which has not yet been determined, is essential for priming activity since its removal erases the primase activity, while maintaining polymerase activity [13]. In this region, C^419^, H^426^, C^446^, and C^451^ residues coordinate with the covalent binding of Zn^2+^ of the Zn-finger domain [12,113]. Some of these residues are mutated in cancer (compiled in Genomic Data Commons Data Portal, GDC [114] and Catalogue of Somatic Mutations in Cancer, COSMIC [115] databases, Table 1), suggesting that these mutations alter the PrimPol function. In fact, specific mutations found in tumors, such as R76H and R76C mutations, likely impair PrimPol activity, as it has been previously shown for the Y100H mutation, which disables the entrance of dNTPs in favor of NTPs [74]. 

Solving the structure of PrimPol has shed light on why this enzyme is able to directly copy some lesions and not others. On one hand, the AEP core of the human PrimPol structure was determined in complex with a DNA template containing 8oxodG, a lesion readable by PrimPol. Moreover, the structure was determined in different experimental conditions: in order to see the error-free and error-prone copying of the lesion, with a primer and an incoming dCTP or dATP in front of 8oxodG [80]. These structures show that the active site of PrimPol has a cleft that can harbor the 8oxodG lesion in the template paired with both dCTP or dATP, without any impediment in both cases [80], in contrast with the previous data showing that human PrimPol prefers dCTP over dATP for tolerating 8oxodG [77,78]. They discussed that this is most likely due to the pair 8oxodG/dCTP, which is more thermodynamically stable than 8oxodG/dATP. Moreover, in order to see 8oxodG elongation once it is copied, the structure was also determined using a primer with a final nucleotide of dCMP or dAMP, which was paired with 8oxodG at the template, and an incoming nucleotide. This structure showed structural changes that make this extension less favorable than elongating the primer termini with dCMP paired with 8oxodG [80], in agreement with the previous studies [77]. Altogether, these structures showed that PrimPol can directly copy 8oxodG as it is able to accommodate this lesion in its active site to be copied and continue with the synthesis afterwards.

On the other hand, the structure of PrimPol showed that the active site cleft cannot harbor UV-induced DNA lesions [107], in agreement with the previous results, which demonstrate that human PrimPol skips this kind of lesions by realigning primers ahead to continue the synthesis (see Figure 3B) [13,76].

As already mentioned, the entire human PrimPol structure, with the Zn finger and RBD in addition to the AEP core, is not yet clear. The Zn-finger region of PrimPol is needed for its primase activity [13], since it binds the 5′NTP or dNTP and contributes to recognizing the cryptic base dG [113]. Therefore, the entire structure of human PrimPol could reveal how this enzyme performs priming. Recently, a new powerful method for protein structure prediction called AlphaFold was developed [118]. We decided to model the entire structure of PrimPol predicted with AlphaFold in complex with a DNA template, an NTP at the 5′position and a dNTP at the 3′position, as well as with the two metal ligands that coordinate these two nucleotides and a cation located at the Zn finger (see details of modeling in Appendix A) as if it were in the moment of priming. The first step of priming is the binding of single-stranded DNA (ssDNA). Our model shows that the C-terminal Zn-finger-containing domain, which includes the key residue for priming R^417^, is predicted to be located close to the DNA template (Figure 7B), in agreement with its function. Priming activity also requires the binding of a 5′nucleotide, but in the model this region is located too far away from the 5′NTP. This can be explained by the fact that the protein structure is predicted without taking DNA and nucleotides into account. It is possible to see that the AEP and the C-terminal domain are connected by a loop, which means that these two domains may have mobility between each other, as previously predicted [113]. Therefore, the C-terminal domain is likely located closer to the 5′ nucleotide when priming. Using this model, we also show the location in the space with respect to the different nucleotides of all the key residues of the RBD and the active site exposed above (Figure 7B). In the latter, it is possible to see that each residue is located in the structure in accordance with each of its functions. Furthermore, it is possible to see that the active site has sufficient room to accommodate an NTP at the 5′ position without any impediment (Figure 4 and Figure 7B). This is something that can be clearly observed in a surface representation of the PrimPol structure model (Figure 7C). As explained earlier, this pocket is a very special feature of the primases.

Proteins that operate in DNA damage responses are thoroughly regulated with different strategies, such as post-translational modifications or binding partner proteins for activity modulation [119]. PrimPol is not an exception to this rule. Regarding the post-translational modifications, PrimPol was found to be polyubiquitinated in cells [120] (depicted in Figure 8, left top panel). This post-translational modification tags the protein for degradation and, therefore, limits PrimPol abundance in cells. Deubiquitinase USP36 actively counteracts PrimPol polyubiquitination, reducing protein turnover [120]. Moreover, USP36 participates in replication stress response in a PrimPol-dependent manner [120] (Depicted in Figure 8, middle panel). The abundancy in cells of PrimPol is also limited by Werner helicase-interacting protein 1 (WRNIP1) [121]. Moreover, it was recently shown that PrimPol is phosphorylated by Polo-like kinase 1 (PLK1) at S^538^, located at its RBD, which is regulated throughout the cell cycle to prevent the aberrant recruitment of PrimPol to DNA [122] (Depicted in Figure 8, left bottom panel). PrimPol is dephosphorylated in response to DNA damage [122] (Depicted in Figure 8, middle panel). The responsible phosphatase remains unknown.

Regarding binding proteins for activity modulation, human PrimPol has several partners. Human PrimPol binds to RPA [15,106] (Depicted in Figure 8, right panel) by its RBD, which is divided in two regions (RBD A and RBD B with D^519^/F^522^ and D^551^/I^554^ as key residues, respectively) [117]. This interaction enhances the primase, polymerase [123] and strand displacement activities of RBD [73]. It has been demonstrated that this interaction takes place upon HU or irradiation (IR) treatment, co-localizing both proteins in the nucleus [15], and that it is essential to recruit PrimPol in the vicinity of ICLs [102]. Moreover, human PrimPol binds in vitro to PolDIP2, enhancing its TLS polymerase activity across 8oxodG [124] and 1,2-intrastrand cisplatin cross-link (1,2-GG CisPt CL) lesions [82] and its strand displacement activity [73]. This interaction is via a flexible loop in PrimPol (aa 200–260, Table 1) and an arginine cluster in PolDIP2 (R^282^ and R^297^) [116]. The biological meaning of the interaction between PrimPol and PolDIP2 has not yet been discovered in vivo. Nevertheless, it is possible to hypothesize that this interaction could take place during replication stress (depicted in Figure 8, right panel), since PolDIP2 is a partner of other TLS-polymerases [125,126] and is recruited to nuclear speckles upon UV irradiation [127]. PrimPol also associates with two mitochondrial proteins, highlighting the relevance of this enzyme in mitochondria: the mitochondrial DNA replicative helicase Twinkle [128], demonstrated in vitro, and *mt*SSB, the mitochondrial homologous to RPA [106], demonstrated in vivo.

## 5. Lesion-Containing Gap Filling after the Repriming of PrimPol 

The repriming action leaves a lesion-containing gap that needs to be filled. After PrimPol action, it is demonstrated that when the gap contains bulky lesions, such as BPDE-DNA adducts, it is filled by template switching in a post-replicative way [101]. In this process, initiated by the exonucleases EXO1 and Mre11 (Figure 9, bottom left panel), the nascent sister chromatid is invaded by the stalled primer, mediated by RAD51, and is normally used as a template by replicative polymerases (Figure 9, bottom right panel), providing an error-free method of damage tolerance [101]. Moreover, it was demonstrated in the S phase that the TLS polymerase complex REV1-Pol𝜁 carried out the filling of PrimPol-dependent gaps with UBC13 and RAD51 as partners [129]. A more direct method, but one that is more error-prone, is to fill the gaps that PrimPol leaves behind, implying the participation of polymerases that perform a trans-lesion synthesis (TLS) across damages (summarized above, [130]). In fact, it seems that the work of polymerases in lesion-containing gaps is their major role [131]. It was demonstrated that the TLS polymerase complex REV1-Pol𝜁 carries out this task in BRCA1/2 deficient cells, a scenario involving the accumulation of DNA gaps due to PrimPol-mediated repriming (Figure 9 top right panel) [132]. This mechanism, dependent on RAD18, takes place during the G2 phase [129]. It might be interesting to know in future research whether other TLS-polymerases are also able to carry out this task, in order to know if each polymerase is necessary, depending on the lesion to be tolerated or the step of the cell cycle. 

## 6. Expression of Human PrimPol in Different Tissues

Altogether, research suggests that PrimPol is a highly relevant enzyme in human cells. To note, this enzyme is highly conserved across organisms and species such as: *Mus musculus* (mice) [12,13], *Gallus gallus* (chicken) [14], the plant *Arabidopsis thaliana* [133], the archaea *Pyrococcus furiosus* [134,135] and the bacteria *Thermus thermophilus* [136,137], highlighting the relevance of this enzyme. Nevertheless, there are organisms such as *Saccharomyces cerevisiae* (yeast) that do not have PrimPol, in which the PrimPol action is carried out by other enzymes or complexes [88].

In order to infer the value of this enzyme in each tissue in humans, whose gene is located at chromosome 4 (GRCh38), we analyzed the PrimPol expression data deposited in GTEx Portal (Figure 10, The Genotype-Tissue Expression (GTEx) Project data portal, Gencode ID ENSG00000164306.10) [138,139]. Remarkably, tissues with the higher PrimPol expression were those that require high turnover rates, such as female reproductive system tissues (uterus, cervix, ovary, and fallopian tube) (Figure 10) [140], and tissues with lower PrimPol expression were those with low turnover rates, such as the liver, pancreas, heart and brain (Figure 10). The same pattern is observed for the expression of PCNA according to the data deposited in GTEx Portal (data not shown). PCNA is a key protein for DNA replication [141,142], and it is commonly used as a marker of cell proliferation [143,144]. These data support the relevance of PrimPol in DNA replication. 

Some interesting exceptions are the tibial nerve, brain cerebellar hemisphere, and cerebellum (Figure 10), which show a relatively high PrimPol expression compared with other tissues, even with low turnover rates [140]. Therefore, it is possible that the PrimPol role in these tissues is replication-independent. In fact, replication-independent roles were shown for bacterial PrimPols. PrimPols from *Marinitoga piezophila* and *Dysgonamonadaceae bacterium*, physically interact with Cas proteins, suggesting a function in CRISPR-Cas adaptation [145]. Moreover, PrimPol from *Thermus thermofilus* [136] plays a role in conjugation [137]. It could be interesting to explore if PrimPol has replication-independent roles in human cells. On the other hand, the stomach tissue, which is the tissue with the highest turnover rate, shows a low expression of PrimPol compared with other tissues (Figure 10). This could mean that the repriming process is carried out by the primosome and not by PrimPol, as happens in organisms that do not possess PrimPol, such as *Saccharomyces cerevisiae* [88], or maybe this pathway is supplied in this tissue by other pathways. The further investigation of the relevance of repriming in different tissues could produce interesting clues regarding the DNA damage tolerance and repair choices in each of them.

## 7. Role of PrimPol in Cancer and Its Potential Therapeutic Implication

The role of replicative [55,146,147] or non-replicative polymerizing [57,146,148,149] enzymes in cancer has been widely studied. It has been reported in breast cancer that tumors with deficient PrimPol expression have a higher mutation burden [150]. Thus, we decided to analyze the data deposited in GDC Portal [114] and COSMIC data base [115] for PrimPol copy number variations (CNV, in GDC) and mutations (in GDC and COSMIC), in order to understand its possible role in other cancer types. We observed that cancers with higher PrimPol alteration rates (CNV Figure 11A, or mutations Figure 11B) were those involving tissues related to the female reproductive system, which are the tissues with higher PrimPol expressions in physiological conditions. Altogether, these data suggest that the perturbation of PrimPol could have an effect on cancer development.

Specifically, we observed cancer types with a recurrent copy number (CN) gain of PrimPol locus (Figure 11C), suggesting that this genomic alteration could confer an advantage to these tumors. In this sense, cancer cells are characterized by an increased genome instability [151] and PrimPol could alleviate an excess of this trait, maintaining genome instability under the critical threshold that compromises cell viability (Figure 12B). Another scenario, in which cancer cells could show high PrimPol activity, is in the presence of inactivating mutations in genes that negatively regulate PrimPol. This is the case for tumors with mutations in BRCA1, BRCA2, or RAD51. These three proteins downregulate PrimPol, and their dysfunction triggers more PrimPol-dependent repriming (Figure 12C) [103,132,152,153]. Nevertheless, in all these scenarios, where PrimPol is overacting, cells accumulate a high number of gaps that need to be filled. If cells are unable to fill all of these gaps, genomic instability will increase. Therefore, the over-repriming action of PrimPol is a double-edged sword, as it can limit fork stalling (Figure 12B) but at the cost of increasing the number of gap-containing lesions to be filled (Figure 12C), which could lead to genomic instability. In this regard, recently published research showed that the homologous recombination of defective cells is more sensitive to complex REV-Pol ζ inhibitors, leading to a gap accumulation [129,132]. Therefore, it is tempting to suggest that PrimPol-over-acting cancer cells could be treated with REV-Pol ζ inhibitors. Moreover, in the hypothetical case of patients with cancer cells that are REV-Pol ζ deficient, the action of PrimPol could be exacerbated by the administration of several cisplatin doses [103]. This would cause a synthetic lethality, in which the gaps generated by PrimPol repriming in response to the action of cisplatin are not filled due to the lack of REV-Pol ζ, triggering an intolerable genomic instability. 

On the other hand, taking into account that several cancer treatments are based in their genotoxicity, PrimPol over-action could be responsible for developed resistance. This could be the situation with several doses of cisplatin [103], since BRCA-deficient cells subjected to this treatment do not show fork degradation due to the over-expression of PrimPol. In this case, it was suggested that it could be possible to eliminate this resistance by targeting PrimPol with inhibitors [102]. In this regard, PrimPol inhibition has already been demonstrated in vitro with different aptamers [154], a kind of molecule used in cancer therapy [155]. In the next few years, we will see if this strategy is therapeutically viable.

Finally, the over-action of PrimPol in cancer cells, as well as cancer cells that have more PrimPol dependency due to the defective status of other genes [150], could be exploited by administrating nucleotide analogs that are already used by PrimPol in vitro in order to generate genotoxicity. Some good examples of this are the anti-cancer drugs, cytarabine (Ara-CTP) and gemcitabine (dFdC) [109], or the nucleoside reverse transcriptase inhibitors (NRTIs) used normally as the anti-virals ddATP, ddCTP, zidovudine triphosphate (AZT-TP), and tenofovir diphosphate CBV-TP [156].

However, there are cancer types with a PrimPol CN loss (Figure 11D) or likely a loss of function mutations, such as those at residue R76 (Table 1). In fact, this residue is a mutation hotspot (Figure 11E). PrimPol alterations in tumors could mimic some features of the PrimPol-deficient phenotype observed in cell models: increased replicative stress and genomic instability compared to WT cells [13,94] (Figure 12D). Therefore, it is tempting to suggest that these patients could benefit from radiotherapy or chemotherapy treatments to which PrimPol-deficient cells have already been shown to exhibit hypersensitivity, such as the interstrand crosslink agents cisplatin [96,102,103] and mitomycin C [102], or even the combined treatment of camptothecin (CPT) or etoposide with olaparib. The rationale for this latter treatment is based on the fact that the poisoning of topoisomerase I by CPT results in PARP-mediated replication fork reversal [157], which could be suppressed by PARP inhibitors, such as olaparib. Therefore, the treatment of PrimPol-deficient cells with both the topoisomerase I poison CPT and the PARP inhibitor olaparib may be highly detrimental to their viability, as these cells have a natural absence of repriming activity, and thus lack an alternative mechanism to face genotoxicity.

In addition, PrimPol could also be an enzyme considered in pharmacogenomics, since its activity or deficiency in patients could be a contraindication for certain types of treatments. This seems to be the case regarding an HIV+ patient treated with tenofovir who harbored the PrimPol mutation D114N [99], a mutation that inactivates the primase activity of PrimPol. PrimPol-deficient cells are hypersensitive to tenofovir treatment, which could explain the patient’s toxicity [99].

## 8. Concluding Remarks

The discovery of PrimPol, the first DNA primase discovered in human cells, was a breakthrough in the DNA replication field [12]. PrimPol provides the option of repriming ahead of obstacles during DNA replication [13], leaving behind a gap that will be filled by template switching [101] or TLS polymerases [129,132] (depicted in Figure 12A). Moreover, in vitro studies uncovered that PrimPol is able to use its TLS abilities when polymerizing [12,13], acting as a TLS DNA polymerase. In vivo, PrimPol DNA primase activity has been demonstrated; however, PrimPol TLS DNA polymerase action has not yet been probed. To carry out its priming activity, PrimPol possesses a Zn-finger domain that plays a key role in this function [113]. Here, we show by structure modeling that this domain is theoretically located close to the template, which is in accordance with the function of this domain since the ssDNA binding is the first step for this activity. 

The relevance of PrimPol to perform an efficient DNA replication, and thereby cell proliferation, becomes evident when noticing that the human tissues with the highest expression of PrimPol are those with higher turnover requirements. On top of that, PrimPol could have an impact in cancer development, progression and resistance to genotoxic agents, and consequently, it is a clear target of future therapies.

Beyond the relevance of PrimPol in DNA replication, the amazing capabilities of PrimPol were studied for biotechnology purposes: PrimPol from *Thermus thermophilus* is used in the method TruePrime for whole-genome amplification from single cells. This method is based on the primase activity of PrimPol, which provides real primers instead of random hexamers, which is the basis of other more common and up-to-date methods [136]. 

Considering the versatility and relevance of PrimPol in DNA replication, new studies that shed light on the biochemistry and in vivo activity of this enzyme will have an impact on cancer therapy and other applications.

## Figures and Tables

**Figure 1 biomolecules-12-00248-f001:**
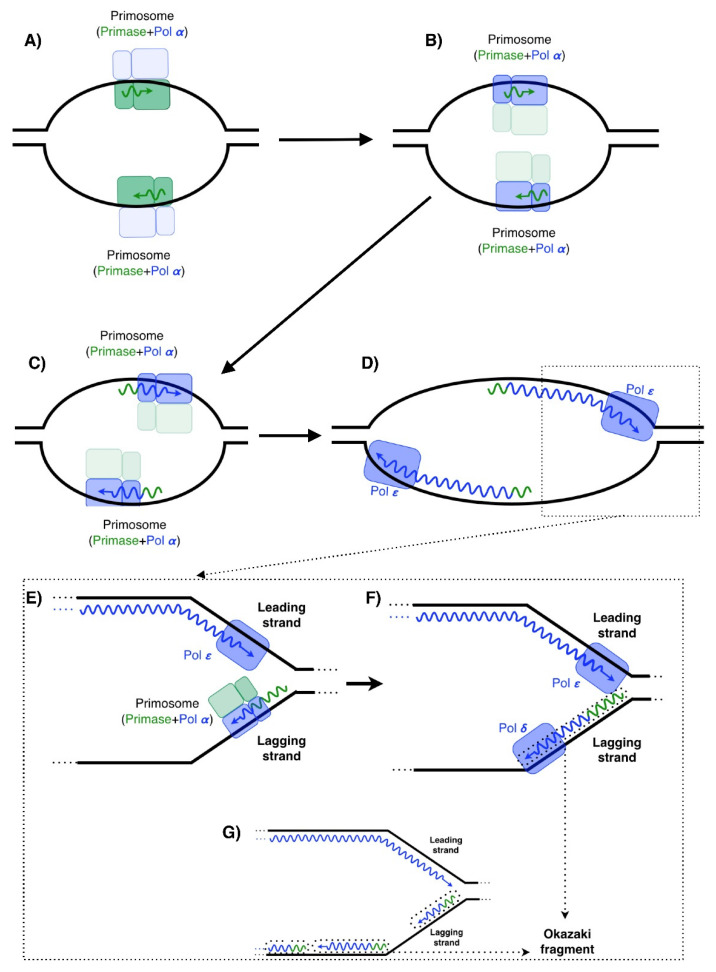
DNA replication in human cells. Leading-strand: (**A**) The synthesis of DNA is initiated on both strands by the primase (composed by the catalytic and regulatory subunits) catalyzing the synthesis of RNA primers of 8–9 nucleotides in length. (**B**) The RNA primer is transferred to the DNA polymerase Pol α (**C**) to elongate it with dNTPs, making primers composed of a 5′ portion of RNA and a 3′ portion made of DNA. Primosome: primase + DNA polymerase α. (**D**) Then, the replicative DNA polymerase Pol ε carries out the following elongation. Lagging-strand was (**E**) synthesized discontinuously in small pieces initiated by the primosome and (**F**) elongated by the replicative DNA polymerase Pol δ. (**G**) Okazaki fragments are synthetized in a recurrent manner during lagging-strand synthesis.

**Figure 2 biomolecules-12-00248-f002:**
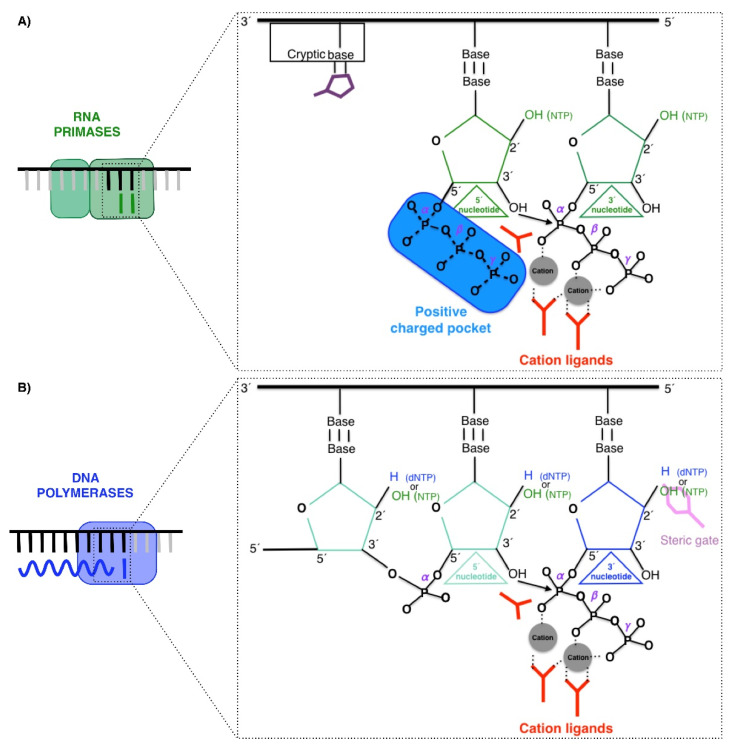
Similarities and differences between RNA primases and DNA polymerases. Primases and polymerases catalyze the same SN2 reaction, in which the 3′OH is the attacking group; the phosphate α of the incoming nucleotide is attacked and the pyrophosphate composed of the phosphate β and γ is the displaced group. However, (**A**) primases possess a pocket with a high positive charge (depicted in blue) to capture the triphosphate of the nucleotide bound at the 5′ site, allowing the use of NTPs as primers. The preference for a priming site is conferred by the stabilization of the 5′ nucleotide by the preceding base in the template (called cryptic base), which is bound by a specific residue (in purple). The positive charge pocket is absent in (**B**) DNA polymerases; therefore, NTPs cannot be used at 5′site and require a pre-existent primer, such as a DNA primer similar to the primer depicted. Moreover, DNA polymerases possess steric gate residues to discriminate the sugar of the 3′-incoming nucleotide. These steric gate residues (depicted in pink) impose steric clashes to the 2′OH group of incoming NTPs, blocking their binding, and thus favoring dNTPs. The tree carboxylates responsible for coordinating the two metal ions are colored in red.

**Figure 3 biomolecules-12-00248-f003:**
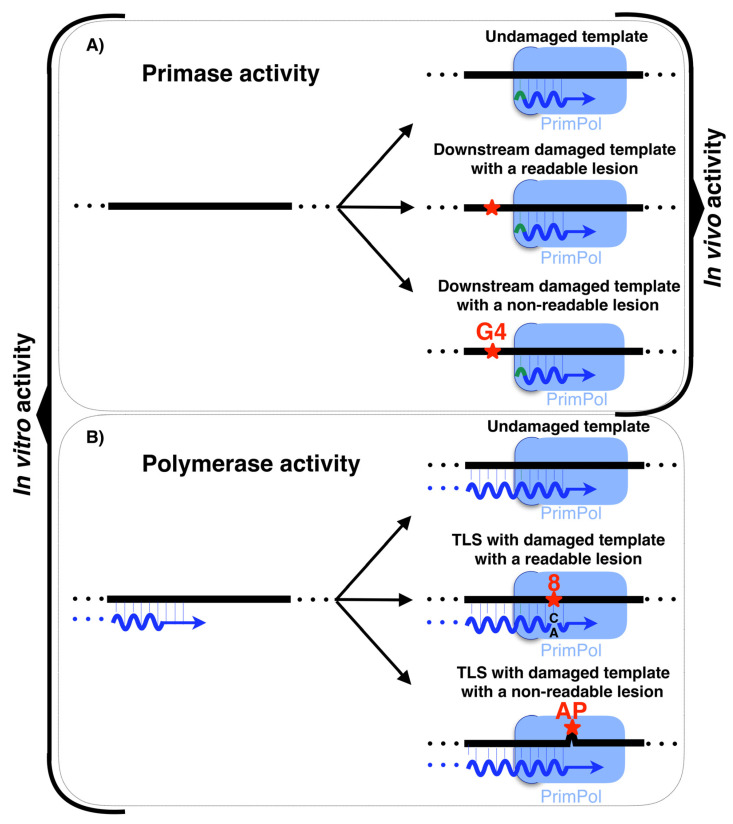
In vitro and in vivo activities of the human PrimPol. (**A**) Primase activity of PrimPol. (**B**) Polymerase activity of PrimPol. Both panels are broken down into polymerization with undamaged templates or damaged templates with both readable and non-readable lesions. Only the primase activity of PrimPol was shown both in vitro and in vivo. Polymerase activities are shown only in vitro and their relevance in vivo remains unclear. dNMPs are colored in blue, NTPs in green and DNA lesions in red. G-quadruplex are depicted as G4 in red, the 8oxodG lesion is depicted as an 8 colored in red (readable lesion), and the abasic site is depicted as AP colored in red (non-readable lesion). Asterisks indicate lesions.

**Figure 4 biomolecules-12-00248-f004:**
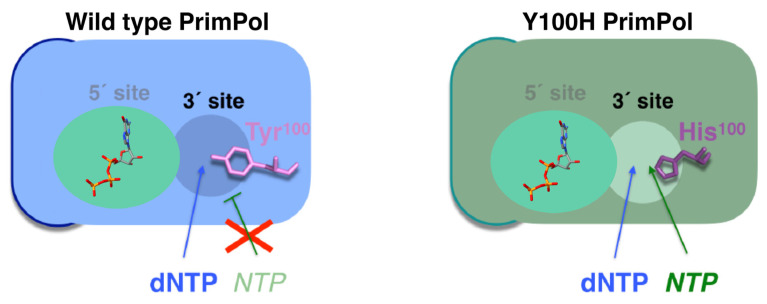
Tyr^100^ of PrimPol is the responsible for blocking NTPs as incoming nucleotides (left), and its change to His allows the entrance and use of these substrates (right). On the other hand, the nucleotide at the 5’ site is most likely an NTP in vivo.

**Figure 5 biomolecules-12-00248-f005:**
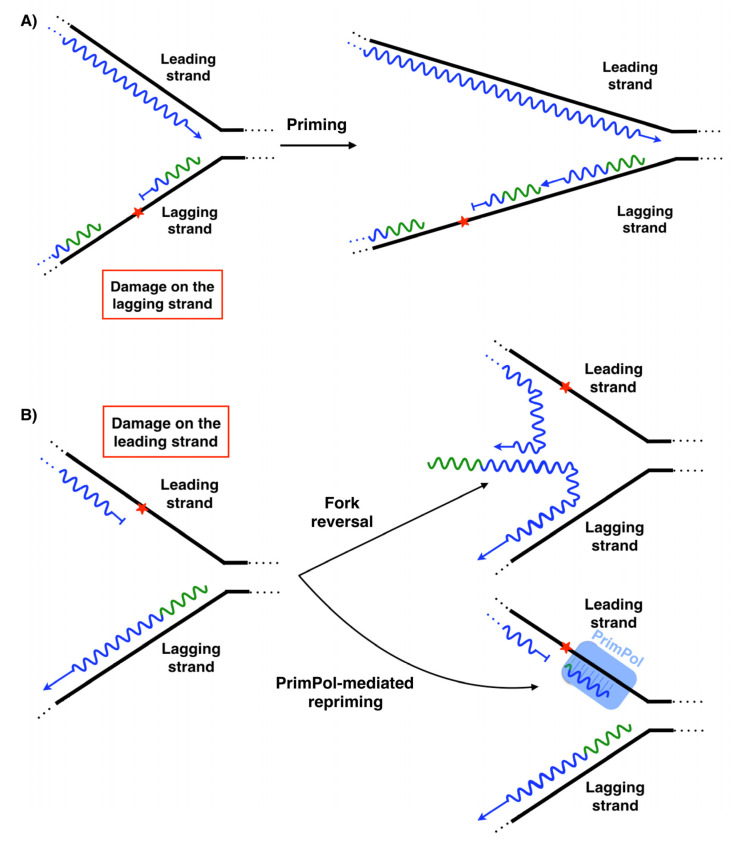
Different ways to avoid DNA blocking obstacles on the lagging and leading strand. (**A**) Obstacles (red asterisk) on the lagging strand are avoided by regular repriming. (**B**) Obstacles (red asterisk) on the leading strand are avoided by different ways: fork reversal and PrimPol-mediated repriming.

**Figure 6 biomolecules-12-00248-f006:**
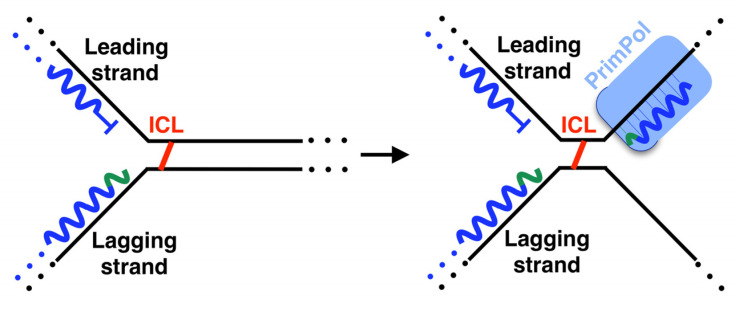
PrimPol facilitates replication traverse of ICLs by priming downstream of the lesion.

**Figure 7 biomolecules-12-00248-f007:**
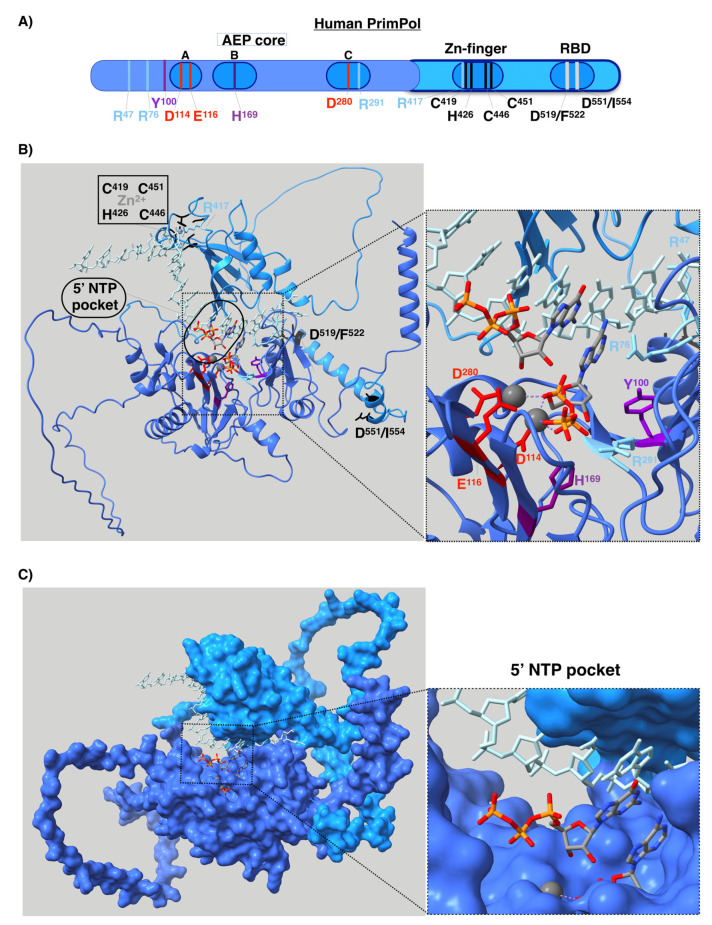
Structure of human PrimPol. (**A**) Linear representation of the structure. AEP domain is colored in blue and C-terminal region in light blue. In motifs (**A**–**C**) (which contain the carboxylates used to coordinate the metals and the key histidine to stabilize the incoming nucleotide), the Zn finger and the RPA binding domain (RBD) are highlighted. (**B**) Model structure obtained from AlphaFold v2.0 protein structure database and modeled in complex with DNA template and 3′dNTP by software fitting with PDB ID 7JKP, and 5′NTP by manual fitting with PDB ID: 7JKP. AEP domain is colored in blue and C-terminal region in light blue, which contains the Zn-finger domain and RPA binding sites, which are colored in black. The template is colored in orange, and 5′NTP and 3′dNTP are colored with CPK code. The key residues are shown in sticks: D^114^, E^116^, and D^280^ colored in red, H^169^ in purple, Y^100^ in dark purple, and R^47^, R^76^, R^291^, and R^417^ in blue. (**C**) Surface representation of the PrimPol structure model showing in detail the 5′NTP pocket.

**Figure 8 biomolecules-12-00248-f008:**
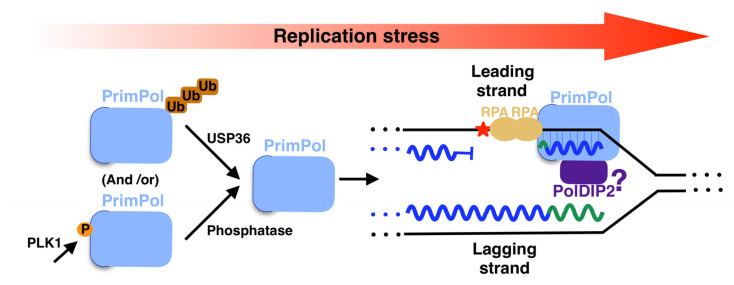
In vivo regulation of PrimPol. when there is no replication stress, PrimPol is polyubiquitinated (left panel). Upon replication stress, PrimPol is deubiquitinated by USP36 (middle panel) and binds at replication fork to RPA and likely to PolDIP2 to carry out its function by repriming ahead of blocking lesions (right panel). Red asterisks indicate lesions.

**Figure 9 biomolecules-12-00248-f009:**
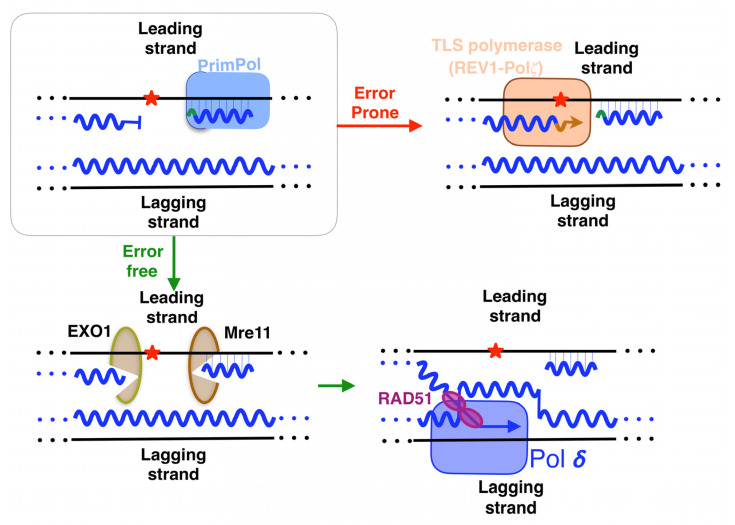
Filling in the gaps that repriming by PrimPol leaves behind. (Bottom panel): template switching which is an error-free way that starts by the resection of both sides of the gap by exonucleases. Then, RAD51 coats a filament of ssDNA, which invades the other duplex to use it as error-free template. Pol δ copy the donor template, and finally, the single strand returns to its starting site to continue the synthesis ahead of the lesion by Pol ε. (Top right panel): translesion synthesis (TLS) made by specialized polymerases, which tolerate the lesions. Red asterisks indicate lesions.

**Figure 10 biomolecules-12-00248-f010:**
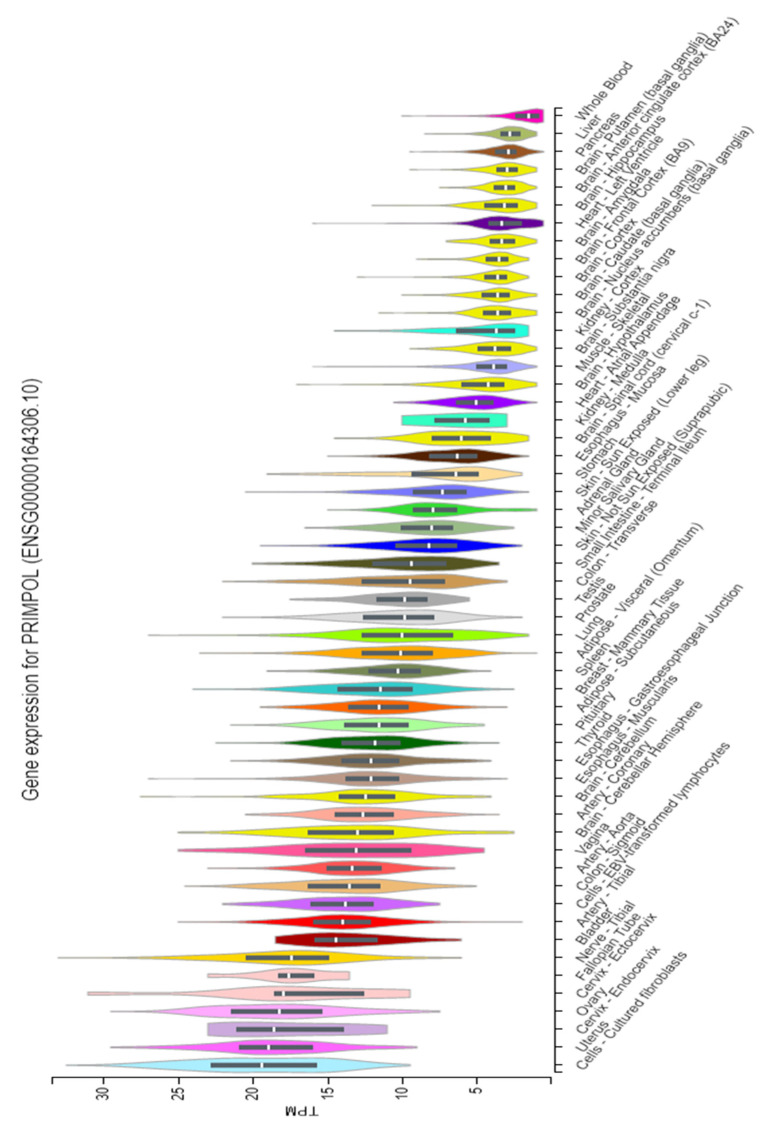
PrimPol expression in different human tissues. Tissues with a higher expression of PrimPol are from the female reproductive system (uterus, cervix, ovary and fallopian tube) and tissues with lower expressions are liver, pancreas, heart and brain. Data and graph were obtained from The Genotype-Tissue Expression (GTEx) Project data portal (ENSG00000164306.109).

**Figure 11 biomolecules-12-00248-f011:**
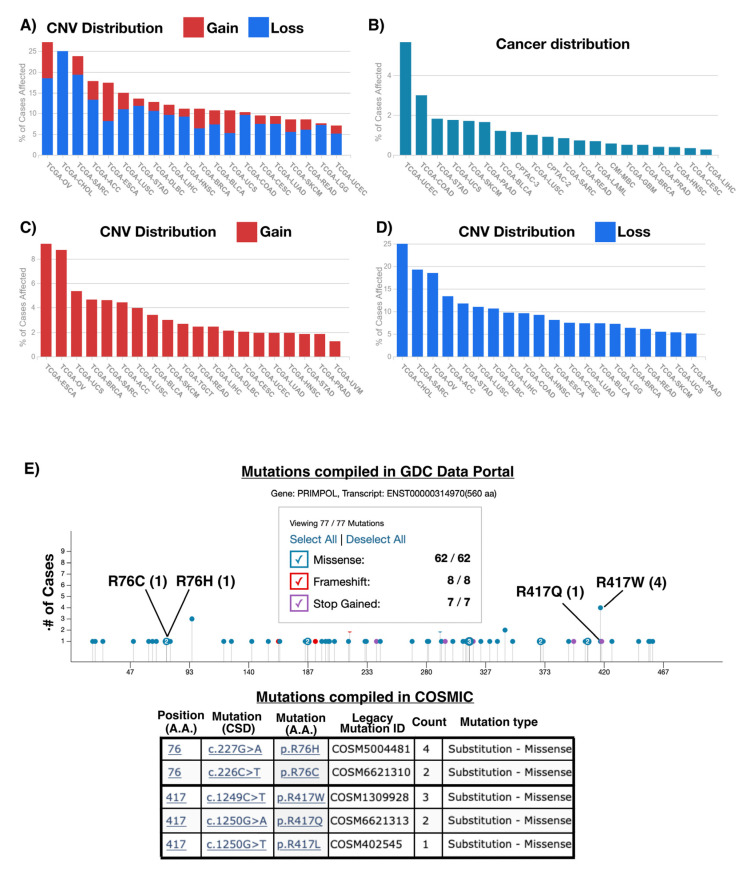
CNV and mutation of PrimPol in different types of cancer. (**A**) The cancer types with the higher percentage of cases affected by either gain or loss of CN are OV, CHOL and SARC. (**B**) The cancer types with the higher frequency of PrimPol mutations are uterine corpus endometrial carcinoma (UCEC), colon adenocarcinoma (COAD), and stomach adenocarcinoma (STAD). (**C**) CN gain of PrimPol is found frequently in esophageal carcinoma (ESCA), ovarian serous cystadenocarcinoma (OV), and uterine carcinosarcoma (UCS). (**D**) CN loss of PrimPol is found frequently in cholangiocarcinoma (CHOL), sarcoma (SARC) and ovarian serous cystadenocarcinoma (OV). (**E**) Missense mutations predicted to impair PrimPol activity were compiled in GDC data portal and COSMIC.

**Figure 12 biomolecules-12-00248-f012:**
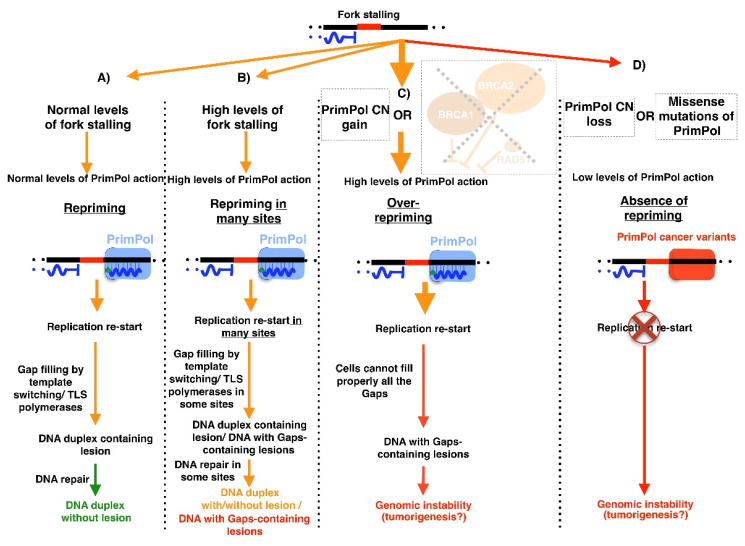
Different genomic situations (**A**–**D**) in normal and cancer cells with a human PrimPol point of view.

**Table 1 biomolecules-12-00248-t001:** Key residues of human PrimPol. Function, activity and cancer-related mutations compiled in COSMIC or GDC databases are indicated. Residues frequently mutated in cancer are indicated by an asterisk. The AEP domain is colored in blue and C-terminal region in light blue.

Residue	Function	Activity Involved [Ref.]	Somatic Mutations in Cancer (COSMIC Database)
R^47^	Contact the DNA template	Primase/polymerase [107,112]	-
*R^76^	Contact the DNA template	Primase/polymerase [107,112]	R76H and R76C
Y^100^	Steric gate (sugar selector)	Primase/polymerase [74]	Y100H
D^114^	Cation ligand (Mn^2+^ or Mg^2+^)	Primase/polymerase [12,79,107]	-
E^116^	Cation ligand (Mn^2+^ or Mg^2+^)	Primase/polymerase [12,79,107]	-
H^169^	Stabilize the 3´ nucleotide	Primase/polymerase [12,107]	-
L^200^-S^260^	PolDIP2 binding	Primase?/polymerase [116]	G201D, E203K, D204G, A208S, A208T, H214Y, P217S, P217L, H218Y, F219L, S220L, Q226L, K232T, M233I, T235R, W243S, T244A, G254W, SS59R
D^280^	Cation ligand	Primase/polymerase [12,79,107]	-
R^291^	Stabilize the 3´ nucleotide	Primase/polymerase [107,111]	R291W
*R^417^	-	-	R417L, R417W and R417Q
C^419^, H^426^,C^446^ and C^451^	Zn^2+^ ligand	Primase [12,113]	H426N and H426R
D^519^/F^522^ and D^551^/I^554^	Binding of RPA	Primase/polymerase [117]	F522V

## Data Availability

Not applicable.

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
