# Peer review of "PrimPol: A Breakthrough among DNA Replication Enzymes and a Potential New Target for Cancer Therapy"

_biomolecules, 2022, doi:10.3390/biom12020248_

Round 1
Reviewer 1 Report
The review from Alberto Diaz-Talavera et al. is an exhaustive compendium of informations on the PrimPol enzyme and it certainly deserves to be published in Biomolecules.
From my point of view, the readers would benefit from a more extended discussion of the “theoretical caveats” that some of the presented models describing the in vivo PrimPol roles imply.
To improve the reading of the manuscript, I suggest that the writing style of several paragraphs is re-visited.
Also, the presence of typos could be carefully addressed before publication.
General Comments
Line 56-268. The authors could consider the possibility to comment that there is a complex between Polepsilon and the MCM-Cdc45-GINS (CMG complex). The CMG complex binds Ctf4 and the pausing complex (Mrc1-Tof1-Csm3).
In this context, the authors should consider the possibility to mention that, up to know, the mechanism of re-priming on the leading strand is not understood.
Is the PrimPol primer placed downstream the MCM complex?
In this case the primer would be displaced by the subsequent advancement of the replisome and PrimPol would need a DNA helicase to unwind the DNA ahead of the MCM and the obstacle.
Is the MCM complex unloaded and re-loaded downstream the PrimPol primer?
Is there uncoupling of the CMG complex from polepsilon when the replisome encounters an obstacle so that the PrimPol primer can be deposited within the two?
Moreover, there are certain lesions/obstacles that can enter the MCM barrel and reach Polepsilon but others cannot. The authors should consider the possibility to comment and discuss that the re-priming on the leading strand is a very complex process. Critical points of leading strand re-priming should be discussed in the review, and it should be stated that the mechanics of the re-priming mediated bypass on the leading strand is not understood. From my point of view the models proposed until now do not explain how this event can occur in the context of the structure and dynamic of the replisome. For example, how can Polepsilon be unloaded from the blocked growing chain (and from the CMG complex) and be re-loaded downstream the block to elongate the PrimPol primer?
Moreover, the CMG complex is tightly associated to Polepsilon and can be loaded on the DNA only one time/cell cycle when the origins of replication are licensed and fired. Because of this, it is difficult to envisage unloading and re-loading events of the CMG-polepsilon complex when the forks face DNA replication obstacles.
The MCM embraces the DNA filament so it cannot be opened, unloaded, and re-loaded outside the context of origin licensing and activation.
My suggestion is that the authors clearly state that, although the PrimPol-mediated re-priming mechanism on the leading strand (theoretically) is a very attractive/convenient/elegant mechanism of fork re-start (and PrimPol action), the enzymology and the mechanics of this proposed process is not understood.
Line 84. The authors could also mention DNA primases utilized by viruses
Figure 2. the authors should describe what cation Ligands are (which amminoacids?) at least in the figure legend
Line 170. Please discuss that errors in the Polalpha DNA initiator fragments can be corrected also by the strand displacement activity of Poldelta (combined with accessory DNA helicases-nucleases) during the processing of the okazaki fragments. Please cite relevant literature.
Line 213. The authors should mention if PrimPol has the positively charged pocked mentioned in figure 1A for other DNA primases. The authors should evaluate the possibility of adding the drawing of that pocket in the 5’ site of PrimPol in figure 4.
Line 284. The authors should consider the possibility to discuss that although the fork reversal enzymes mentioned in the review have in vitro biochemical activities (on purified DNA substrates), that can account for their proposed fork reversal activity in vivo, the enzymology of the in vivo fork reversal process in the context of the replisome structure and dynamic is completely unknown. For example, to perform the fork reversal actions depicted in figure 5b the replisome should be back tracked/unloaded? from the nascent strands that will undergo fork reversion and subsequently re-loaded? (or pushed back to its initial position) to allow DNA replication fork re-start. The enzymology of those fork reversal-related processes is not known. Moreover, from the main text of the manuscript, figure legend and figure 5b, it is not clear (to me) which is the role of PrimPol in the fork reversal-mediated mechanism of lesion bypass depicted. If PrimPol does not have any role in the fork reversal-mediated damage bypass mechanism described/depicted, the authors should mention it in the main text. Regarding the PrimPol mediated leading strand re-priming event reported in Figure 5b, please see the general considerations reported in the paragraph here above. I think the authors should consider the possibility of removing the word “unscheduled” from figure 5b, figure 5b legend and from the main text. The reason of this is that some of those “unscheduled” re-priming events could lead to an error free DNA damage bypass that could be extremely beneficial for the cells. Moreover, up to know, the extent of utilization of the error-free damage bypass through fork reversal towards error free damage bypass through re-priming is not known. The error-free damage bypass through re-priming does not imply that the fork facing the obstacle is backtracked/remodeled/unloaded/reloaded/disassembled/reloaded etc etc.
Line 305. In the model depicted in figure 6 it is not clear (to me) how the MCM associated to the left coming fork would pass through the inter-strand DNA cross link. The authors should report a graphic model that shows how the left coming fork can engage the PrimPol primer deposited downstream the DNA inter-strand cross link and be re-started. Several models have been proposed in the literature.
Line 390. Please describe to the readers what are the A and B domains in the PrimPol structural representation in figure 7A. At least in the figure legend.
Minor Comments
Line 17. “catalyzes” may be replaced by “synthesizes”
Line 68. Remove “and”
Line 88. Substitute “the synthesis de novo” with “de novo synthesis”
Line 92. Check “possesses”
Line 133. Please add “DNA” before “poly-“
Line 139. Convert “polymerases” to “DNA polymerases”
Line 156. Convert “its” to “their”
Line 159-166. Please modify the “writing style” of the sentences
Line 175. Replace “polymerases” with “DNA polymerases”
Line 175-182. Please modify the “writing style” of the sentences
Line 183-185. Please mention PolQ
Line 231. Please check “cannot read directly”
Line 242. Please replace “also is a TLS” with “is also a TLS”
Line 256. Please replace “context” with “contexts”
Line 264-265. Please re-shape the sentence
Line 326. Please re-shape the sentence
Line 330. Check “that has been studied in vitro”
Line 334. “erases” or “erase”?
Line 337. “to” or “for”?
Line 338. “to alanine”
Line 340. “their” “groups”
Line 341. “Their” “to”
Line 344. “The structure of which”?
Line 349. “to”
Line 358-359. Please re-shape the sentence
Line 360. Remove “situations” and replace with “experimental conditions”?
Line 421. Remove “one”
Line 430. “to be polyubiquitinated”
Line 446. “takes”
Line 471. Please remove “which suggest that it is via template switching”
Line 497. Please re-shape the sentence
Line 528-530. Please re-shape the sentence
Line 566. “their”?
Line 582-583. Please re-shape the sentence
Line 590. Check “that It have already been shown”
Line 625. Check “has supposed”
Line 633. “Template” instead of “Patch”?
Line 637. “Or” or “for”?
Line 638. Remove “also”?
Line 647. “In” or “on the”?
Line 647. “providing” or “which provides”?
Line 648. “to this day” or “utilized today”?
Author Response
Dear Reviewer 1,
Please, find attached the file with our point by point reply to your comments.
Best regard,

Reviewer 2 Report
Review for “PrimPol: a breakthrough among DNA replication enzymes with 2 potential as target for cancer treatment”
This is a comprehensive review of human PrimPol function in translesion synthesis and DNA replication. The authors cover major milestones in biochemical, in vitro, and in vivo PrimPol research, highlighting functions and mutations with a potential to impact cancer biology.
Strengths: comprehensive coverage of major topics, use of figures, and overall structure of manuscript.
Weaknesses: grammar, repetitive sentence structure, long and confusing sentences in some areas.
List:
- line 1: “with 2 potential as target for cancer treatment” does not make sense.
PrimPol: a breakthrough among DNA replication enzymes with 2 potential as target for cancer treatment — PrimPol: a breakthrough among DNA replication enzymes as a target for cancer therapeutics (?)
-line 16: In vitro – italicize
-line 18-19: However, the absence of evidences of PrimPol polymerase activity in vivo suggests ... absence of evidences — lack of evidence
-line 19: in vivo – italicize
-line 67-68: “(g) Okazaki fragments and are synthetized in a recurrent manner in the lagging-strand synthesis.”
— fragments are synthesized in a recurrent manner in lagging strand synthesis
-line 102: “at both leading and lagging strand,” — at both leading and lagging strands,
-line 131-135: One long and confusing sentence
-line 144: “…potential hazard, as dNMPs embedded in DNA can alter the helix and impair its proper functions [33,34].” — NMPs instead of dNMPs here?
-line 159-161: “This fact means that only are capable to make the Watson and Crick base pairs but not others, making a high fidelity synthesis, but incapable to carry out non-canonical reactions [reviewed in 27].” — something missing in this sentence only what are capable?
-line 175-6: “Polymerases that are able to make trans-lesion synthesis (TLS) have a laxer active site to harbor inside the lesion on the template and the incoming nucleotide.” — Polymerases that are able to make trans-lesion synthesis (TLS) have a laxer active. [last part confusing and stated nicely in following sentences]
-line 225: reading directly — directly reading
-line 240: Summarizing, — In summary,
-line 249-250: “…polymerases [64], but, in addition to that, it has been shown that PrimPol also requires this feature…” — “polymerases [64], however PrimPol also requires this feature…
-line 245-257: confusing paragraph structure — 1 paragraph is 2 sentences, and the next a single sentence.
-line 274-5: “Therefore, obstacles on the lagging strand cannot stall the replication fork.” — not strictly true. Yes they can stall a fork, but the consequence is minimal in comparison to stalling on the leading strand… restate?
-line 297: “obstacles as UV lesions [13, 87],” — obstacles such as UV lesions [13, 87],
-line 308-9: “…repriming as an adaptive response to multiple cisplatin doses treatment [95] (Figure 5b).” — …repriming as an adaptive response to treatment with multiple doses of cisplatin [95] (Figure 5b).
-line 315: understating — understanding
-line 338: “R291 [103] for alanine” — R291 [103] to alanine
-line 363: “active site of PrimPol have a cleft” — active site of PrimPol has a cleft
-line 416: close to DNA template — close to the DNA template
-line 417: “Priming activity also requires the binding of 5’nucleotide, but in contrast, in the model” — Priming activity also requires the binding of 5’nucleotide, but in the model
-line 432: abundancy — abundance
-line 435: abundancy — abundance
-line 478: on RAD18, take place — on RAD18, takes place
-line 497: “Altogether suggest that PrimPol is an enzyme with high relevance in human cells.” — sentence fragment
-line 522: “[129] play a role in conjugation [130].” — [129] plays a role in conjugation [130].
-line 563: “Other scenario in which” — Another scenario in which
-line 567-8: “Nevertheless, in all these scenarios, where Prim-Pol is over-acting, it is accumulated a high number of gaps to be filled.” — Nevertheless, in all these scenarios, where Prim-Pol is over-acting, cells have accumulated a high number of gaps to be filled.
-line 568-9: “If cells cannot manage to fill all these gaps could finally increase genomic instability.” — sentence fragment
-line 576-9: confusing and fragmented
-line 581: “could be the responsible of developing resistance” — could be responsible for developing resistance
-line 585-7: In fact, it has been already tested in vitro different ap-tamers, a kind of molecules used in cancer therapy [reviewed in 143], that inhibits PrimPol 586 activity [144]. — confusing
-line 605-9: long sentence, confusing to follow
-line 611-3: “…in pharmacogenomics in near future, since its defective performance in patients could be over harmful for different kind of treatments.” — in pharmacogenomics, since its activity in patients could be harmful for different kind of treatments.
-line 625: supposed — ?
-line 639: “…The relevance of PrimPol to perform a correct DNA replication and thereby” — The relevance of PrimPol to perform correct (efficient / faithful may be more appropriate?) DNA replication and thereby…
-line 645: Thermus thermophilus — italicize
-line 647: is based in primase — is based on primase
Author Response
Dear Reviewer 2,
Please, find attached the file with our point by point reply to your comments.
Best regard,
Alberto Díaz-Talavera

Reviewer 3 Report
This is generally well-written and comprehensive review on the structure and functions of PrimPol primase-polymerase enzyme, a recently discovered critical regulator of DNA damage tolerance in human cells. The figures are nicely designed and complement the text well. The review starts by briefly presenting the eukaryotic DNA replication machinery, highlighting structural and functional differences between RNA primases, DNA polymerases and PrimPol itself. The roles of PrimPol in the response to DNA replication stress and DNA damage tolerance as well as its relationship with replication fork reversal is then discussed. The structure and regulation of PrimPol is examined at the amino acid level and cancer-associated mutations that impact its various activities are presented in a convenient table. Alphafold structure predictions and PDB deposited structures were also used to fit a DNA template along with 5’ NTP and incoming 3’dNTP into a “full length” enzyme and this is used to discuss the relevance of specific amino acids for the priming function of PrimPol. The pathways used for gap filling post-repriming by Primpol are presented along with the expression pattern and potential clinical relevance of PrimPol targeting. Finally, a model of the consequences of PrimPol misregulation is proposed.
The manuscript is scientifically sound and appropriately references prior work on PrimPol. Some english editing is required as there are many typos and grammatical errors that could be removed to make the text easier to read (see non-exhaustive list below). A few minor points should also be addressed before the manuscript is suitable for publication.
Minor points :
- On page 16, the expression levels of PrimPol across tissues is presented and the fact that seemingly more proliferative tissues express more PrimPol is proposed to be in accordance with the role of PrimPol in DNA replication. Is the tissue-specific expression pattern of PrimPol similar to that of other DNA replication factors ? Otherwise, I would remove or tone down the sentence on lines 511-513.
- At lines 286-287, fork reversal is presented and it is stated that this process is catalyzed by ZRANB3, HLTF and SMARCAL1, however, other fork remodelers exist and have demonstrated in vivo roles in this process and/or have been shown to promote reversal in vitro (see for instance FBH1 (Fugger et al. Cell Reports 2015). Thus, these other demonstrated or putative remodelers should also be discussed.
- At lines 573-575, the authors state that homologous recombination defective cells may be more sensitive to REV1-pol zeta inhibitors. This has been recently demonstrated in Taglialatela et al. Mol Cell 2021 and Tirman et al. Mol Cell 2021. These publications are discussed elsewhere in the manuscript and these results should be discussed in this part of the text as well.
- At some instances in the text the word catalyzes is used to discuss the fact that PrimPol synthesizes primers at or ahead of the lesion. The word produces or synthesizes should be used instead of catalyzes (e.g. line 238 or 242.
- I fail to see how treating patients with UV light would help treat their cancers. The discussion around lines 602-604 should focus on chemotherapeutic treatments.
- In vitro and in vivo and all other latin expression should be in italics throughout the text.
- GAP should not be in capital letters (gap or gaps).
Typos and grammatical errors
Line 19. Evidence
Line 38 in the 5’-3’ direction which implies…
Line 53 a solution is required to continue synthesis, repriming being one of them.
Line 61 initiated on both strands
Line 62 The RNA primer
Line 68 Okazaki fragments are synthesized in a recurrent manner during lagging-strand…
Line 79. Nevertheless, these three enzymes have the following particularities (differences between them are implied by the particularities).
Line 88. de novo synthesis of small…
Line 96. Neighboring base
Line 102 on both leading and lagging strands and in a recurrent manner on lagging strands for each Okazaki…
Line 121 allowing the use of
Line 125 such as a DNA
Lines 137 so-called
Line 147 tyrosine
Line 156 their structure
Line 159. that they are only capable
Line 162. accurate at copying
Line 177. are capable of carrying out this task
Line 188. The most recently discovered…….. is Primpos, a novel primase-polymerase…
Line 203. panels are broken-down into ???
Line 211. NTPs
Line 225. act as a TLS
Line 231 cannot be read
Line 252 published
Line 256 contexts
Line 344 The C-terminus of the protein, the structure of which has not yet been solved… activity while maintaining polymerase activity
Line 359 structure has been solved in complex with a DNA template
Line 361 and elsewhere. error-free and error-prone
Line 367 thermodynamically
Line 370 that make this extension less
Line 373 able to accommodate this lesion
Line 405 The Zn-finger
Line 409 has been developed
Line 411 position
Line 412 coordinate
Line 414 single-stranded DNA
Line 417 binding of a 5’nucleotide
Line 440 The responsible phosphatase remains unknown.
Line 478 takes place
Line 497 Altogether, research suggests that
Line 516 show
Line 521 CRISPR
Line 522 plays a role
Line 522 has replication-independent roles in human cells
Line 526 carried out by the primosome…, as happens in organisms that do not…
Line 529 could produce interesting clues
Line 537 that tumors with deficient PrimPol expression
Line 543 conditions
Line 566 their dysfunction
Line 573 gap filling
Line 582 BRCA-deficient cells subjected to this treatment do not show fork degradation due to the…
The sentence at lines 585-587 needs to be rewritten to make it more clear.
Line 590. Administering nucleotide analogues that have already been shown to be used by PrimPol in vitro…
Line 600 patients could benefit from radiotherapy…
Line 606 could be highly detrimental for PrimPol-deficient cells
Line 612. could be a contraindication for certain types of treatments.
Line 625. The discovery of PrimPol, the first DNA primase discovered in human cells, was a breakthrough in the DNA replication field. PrimPol provides…
Line 632 other lesions are found
Line 638 with its
Line 647 based on primase
Author Response
Dear Reviewer 3,
Please, find attached the file with our point by point reply to your comments.
Best regard,
Alberto Díaz-Talavera
